# OPENPATCH: A 3D PATCHWORK FOR OUT-OF-DISTRIBUTION DETECTION

## ABSTRACT

Moving deep learning models from the laboratory setting to the open world entails preparing them to handle unforeseen conditions. In several applications the occurrence of novel classes during deployment poses a significant threat, thus it is crucial to effectively detect them. Ideally, this skill should be used when needed without requiring any further computational training effort at every new task.

Out-of-distribution detection has attracted significant attention in the last years, however the majority of the studies deal with 2D images ignoring the inherent 3D nature of the real-world and often confusing between domain and semantic novelty. In this work, we focus on the latter, considering the objects' geometric structure captured by 3D point clouds regardless of the specific domain.

We advance the field by introducing OpenPatch that builds on a large pre-trained model and simply extracts from its intermediate features a set of patch representations that describe each known class. For any new sample, we obtain a novelty score by evaluating whether it can be recomposed mainly by patches of a single known class or rather via the contribution of multiple classes. We present an extensive experimental evaluation of our approach for the task of semantic novelty detection on real-world point cloud samples when the reference known data are synthetic. We demonstrate that OpenPatch excels in both the full and few-shot known sample scenarios, showcasing its robustness across varying pre-training objectives and network backbones. The inherent training-free nature of our method allows for its immediate application to a wide array of real-world tasks, offering a compelling advantage over approaches that need expensive retraining efforts.

## 1 INTRODUCTION

Thanks to recent advancements, deep learning has become a universal tool for achieving automation in various fields, ranging from industrial production processes to driverless vehicles. This success builds on the deep learning models' ability to learn and encode the data distribution experienced during training, which clarifies the reason for the current gold rush toward larger and larger sources of data as well as models: the goal is capturing the variability of the real-world and reducing the risk of facing unknown scenarios at test time. Still, such a strategy does not define a sustainable path for future developments: there will always be a knowledge cutoff date for the models, so they will inevitably encounter something new once deployed in unconstrained conditions. Smarter solutions consist of providing the models with the ability to manage novelty by design, which is the task at the core of the *Out-Of-Distribution* (OOD) detection literature. Research in this field has attracted a lot of attention in the last years, but mainly for 2D data types Hendrycks & Gimpel (2017),Liang et al. (2018),Huang et al. (2021),Liu et al. (2020),Sun et al. (2021), largely neglecting 3D information.

Existing OOD detection techniques need a training phase on a sizable support set of nominal data (*i.e.* ID samples, known categories) to let the model gain knowledge on the concept of *normality* for the task at hand. Hence, none of them can be considered a plug-and-play approach for onboard systems deployed in real-world scenarios, with computational constraints that may prevent any learning stage and only a limited amount of available synthetic ID samples. This is a typical condition for industrial robotics applications where the agents are supplied with only a few clean 3D object templates to use as a reference and they should be able to mitigate potential operational hazards due to novel unexpected objects.

The recently presented testbed 3DOS for 3D Open-Set recognition Alliegro et al. (2022) came with an extensive benchmark of OOD methods originally developed for 2D data that do not necessarily transfer their performance when re-casted to work on 3D data. A common practice in 2D consists of leveraging large pre-trained models and fine-tuning them on the ID support set. However, this process may be suboptimal mainly for two reasons. On one side fine-tuning can lead to catastrophic forgetting, reducing the generalization ability of the original model rather than supporting novelty recognition Wortsman et al. (2022); Kumar et al. (2022). On the other, it requires a learning phase every time the task changes or the nominal support set is updated.

Understanding which is the best way to exploit the rich embedding space learned by large models without fine-tuning is a recent research direction that is attracting the community working on OOD detection with promising results Cappio Borlino et al. (2022); Ming et al. (2022). When moving from 2D to 3D data this strategy sounds well suited as the costs of data collection and model training increase with data dimensionality. Only in the last months, the introduction of massive single and multi-modal 3D datasets have started to change the 3D research landscape, providing the opportunity to extend the analysis of training free strategies for OOD detection on 3D data Deitke et al. (2023); Liu et al. (2023).

With this work we introduce OpenPatch, an approach to extract from the intermediate features of large 3D pre-trained models a patch representation able to capture local and global characteristics that make ID samples of known object categories easily distinguishable from OOD data belonging to unknown object classes. We focus on semantic novelty detection regardless of the data domain as the ID support set is composed of synthetic data while the test samples are real-world point clouds. OpenPatch does not need any tailored learning phase which makes it directly deployable in the open world and suitable for embedded systems.

Inspired by Roth et al. (2022), we consider features extracted at intermediate layers of a deep 3D point cloud pre-trained backbone, which describe object patches that are both semantically relevant (*i.e.* a *leg* of a table), and geometrically significant (*i.e.*. a *cylinder*). Given a new test sample, each of its patches is compared with those collected from the known categories to search for the most similar one. For each best-matching ID patch, we collect both its distance from the test one and its semantic label. We propose estimating a test sample's novelty using both its patches' known class assignments and the corresponding distances from nominal data. The distance alone serves as an indicator of the patch anomaly for each test patch. The sample is unquestionably novel if a sizable portion of its patches is unknown. However, even if every patch appears to be rather typical, the sample may still be new. This is the case when class assignments exhibit high entropy, highlighting the fact that a sample can be reconstructed only through a composition of patches from different known classes, and thus it cannot be assigned confidently to any of them.

OpenPatch outperforms competitors and its results are very promising even in data-constrained scenarios. Through experiments with different backbones and pre-training objectives we showcase the high sample efficiency of OpenPatch and its suitability for industrial OOD detection thanks to its inherent resilience to domain bias and no need for retraining at different tasks.

## 2 RELATED WORKS

Out-Of-Distribution detection is an umbrella term for many subcategories of methods designed to identify novelty at inference time. Part of the differences among these categories originates from the source of novelty (*i.e.* due to covariate or semantic shift) while others relate to the exact experimental setting. The basic OOD framework consists in a simple binary task that separates *known* samples belonging to the same distribution experienced at training time, from *unknown* samples. However, if the ID data is structured in multiple classes, one might want to keep the ability to discriminate them while rejecting novelty. This goes under the name of *Open Set Recognition*. Finally, the focus of *Anomaly Detection* is on locating abnormal parts within an instance. In industrial applications this means training a one-class model able to spot possible components (*e.g.* defective parts) that deviate from the learned *normality*, and the model has to be re-trained for each different class.

In this work we are interested in semantic novelty detection, thus we overview those methods in the OOD literature that can be used with the binary objective of recognizing whether a new sample belongs to one of the known classes or not, neglecting domain or style variations. A simple strategy

consists of relying on the Maximum Softmax Prediction (MSP) of a classifier trained on the known classes Hendrycks & Gimpel (2017). Other approaches have followed the same *post-hoc paradigm* exploiting a classifier output by introducing temperature scaling to reduce overconfidence Liang et al. (2018), energy scores that estimate the probability density of the input Liu et al. (2020), leveraging the norm of the network gradients Huang et al. (2021), or rectifying the network activations Sun et al. (2021). The *outlier exposure* methods Hendrycks et al. (2019) assume the availability of either real or synthetic OOD examples during training but present limited generalization abilities. *Density and reconstruction methods* explicitly model the distribution of known data. This can involve learning a generative model for input reconstruction Abati et al. (2019) or exploiting a likelihood regret strategy Xiao et al. (2020). *Distance-based methods* exploit a learned feature embedding and evaluate sample distances by using the $L^2$ norm Sun et al. (2022), layer-wise Mahalanobis Lee et al. (2018) or similarity metrics based on Gram Sastry & Oore (2020) matrices.

All these methods have been designed mainly with 2D images in mind, while the research on OOD detection on 3D data is just in its infancy and deserves much more attention Masuda et al. (2021); Bhardwaj et al. (2021); Cen et al. (2021); Wong et al. (2019); Riz et al. (2023). This has been clearly pointed out in Alliegro et al. (2022), a very recent work that, besides introducing a testbed for 3D OOD and Open Set problems with different settings of increasing difficulty, has shown with a comprehensive benchmark the mild effectiveness of 2D methods for those 3D tasks.

Model computational cost is a key aspect when dealing with 3D data, thus while designing an OOD solution to be applied on point clouds, it is natural to target approaches that re-use pre-trained models and minimize the learning effort on the ID samples. Still, this effort is mandatory for most of the existing approaches which makes them unsuitable. Only a few recent works have started to propose fine-tuning-free OOD strategies in the 2D literature Cappio Borlino et al. (2022); Ming et al. (2022). This logic has been also applied for 2D anomaly detection with successful results Cohen & Hoshen (2020); Defard et al. (2021); Roth et al. (2022).

## 3 METHOD

Given a support dataset $\mathcal{S} = \{\boldsymbol{x}_i^s, y_i^s\}_{i=1,...N}$ composed of 3D point clouds $\boldsymbol{x}_i$ and the corresponding labels $y_i \in \{1, \ldots, |\mathcal{Y}_s|\}$, the goal of a semantic novelty detection method is to identify whether a new point cloud from the test set $\mathcal{T} = \{\boldsymbol{x}_j^t\}_{j=1,...,M}$ belongs to one of the support classes or not. In other words, the support and the test class sets are only partially overlapping with more classes in the latter $\mathcal{Y}_s \subset \mathcal{Y}_t$. The classes in $\mathcal{Y}_s$ are indicated as *known*, whereas the test classes not appearing in the support set $\mathcal{Y}_{t \setminus s}$ are *unknown*.

Rather than training a model on $\mathcal{S}$ as most of the existing OOD detection approaches would do, we propose to use a pre-trained deep neural network and obtain from it a representation that allows for simple sample comparison. Our method is called OpenPatch, and it has three major components summarized in Fig. 1. The *patch feature extractor* is the procedure that extracts a patch-based representation from nominal samples of the support set. Specifically, we extract features from a mid-level layer of a frozen deep hierarchical neural network in order to obtain patch embeddings that are both semantically and geometrically significant. As second stage, the obtained patch embeddings are organized into class-specific *memory banks*, which are then subsampled to reduce the computational overhead. Thus the output is a feature collection that encodes the concept of normality for the task at hand. Finally, at inference time each test sample undergoes the same patch feature extraction step. We implement a *scoring function* that provides a normality value $\sigma$ on the basis of the nearest neighbor assignment of each patch of the test to one of the support classes and the corresponding distance. The obtained score reflects the confidence with which the sample is assigned to the known classes. In the following subsections, we describe each of these components in more detail.

### 3.1 PATCH FEATURE EXTRACTOR

Convolutional based 3D learning architectures encode point clouds hierarchically through their internal layers. Local features capturing detailed geometric structures from small neighborhoods are subsequently grouped into larger units to generate higher-level features. As the network depth increases, the receptive field of the convolutions expands, allowing deeper layers to encode highly semantic information while modeling increasing portions of the input shape.

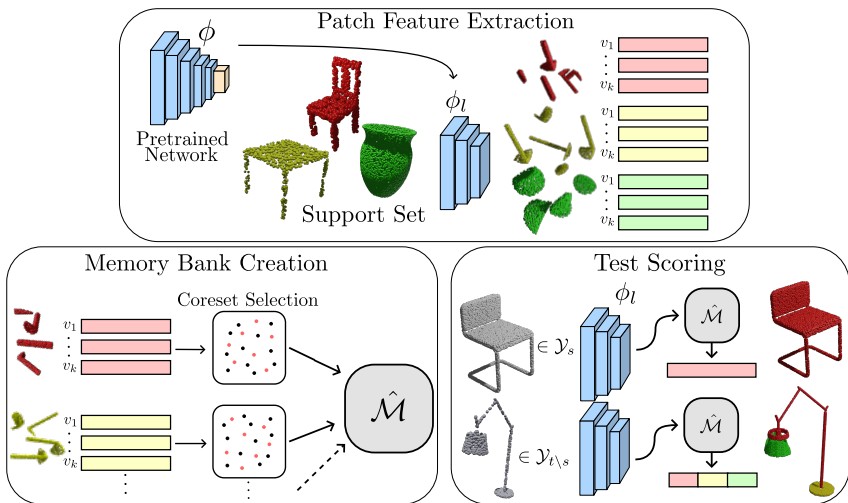

Figure 1: Summary of our approach: OpenPatch uses a model pre-trained on a large scale dataset to extract semantically and geometrically relevant patch embeddings from known samples. These embeddings are used to create a memory bank encoding the task *normality*, which is later used for test sample scoring.

Given an input point cloud $\boldsymbol{x}$ and a 3D convolutional network $\phi$, the feature map in output from its $l$-th layer will be $\phi_l(\boldsymbol{x}) \in \mathbb{R}^{P_l \times C_l}$. This tensor can be read as a set $\{\boldsymbol{v}_{l,k}\}_{k=1,...,P_l}$ of $P_l$ feature vectors each of dimension $C_l$, where the latter value is the number of output channels at the $l$-th layer. Depending on the exact value of $l$, each of these vectors encodes information of a different sized 3D shape portion, so we simply indicate the vectors as *patch embeddings*. The information captured for each patch depends on the specific backbone architecture. Given the inherent complexity of 3D data, it becomes crucial to extract representations that are robust to both rotation and translation of the input point cloud. Although tailored data augmentations are often employed during training to tackle this issue, there are situations where these techniques may fall short. To overcome this limitation, we conduct experiments employing a diverse range of architectures, we mainly explore PointNet++ Qi et al. (2017) but also EPN Chen et al. (2021), an SE(3) equivariant feature encoder. For *PointNet++* Qi et al. (2017) we use the multi-scale grouping classification backbone and extract patch embeddings after the second Set Abstraction (SA) layer. The number $P_l$ of vectors obtained from each input sample is equal to the number of FPS points at the chosen Set Abstraction layer. *EPN* Chen et al. (2021) uses a point convolutional operator that operates on a discretized space of SO(3) rotations. Consequently, each convolutional layer produces a feature map of size $(P_l \times R \times C_l)$, where $P_l$ represents the number of FPS points at that particular layer, $R$ denotes the fixed number of explicit rotations, and $C_l$ represents the number of output channels. To obtain a rotation invariant patch embedding, we employ a symmetric max function that aggregates information across the rotation dimension $R$. This approach is applied individually for each $P_l$ value, meaning for each patch embedding, to ensure that the most salient cues within the discretized set of rotations are preserved.

## 3.2 MEMORY BANKS AND SUBSAMPLING

Starting from the mentioned networks pre-trained on a large dataset, we use their learned representation to describe the samples of the support set $\mathcal{S}$ via patch embeddings. In this process, each obtained feature vector $\boldsymbol{v}_k$ is paired with the label $y_k$ of the point cloud from which it has been extracted: supposing to fix the layer $l$ and to drop this index to simplify the notation, from the whole support set we get $\{\boldsymbol{v}_k, y_k\}_{k=1,...,P_l \times N}$. So the final collection of all the patch embeddings can be split into class-specific memory banks $\mathcal{M}_{y=1}^{|\mathcal{Y}^s|}$.

The number of patch embeddings in each memory bank is determined by the number of support set samples belonging to each class and by the extraction layer $l$. The banks' cardinality may quickly increase, significantly impacting the computational cost of the method. To mitigate this effect and

address redundancy we adopt a greedy *coreset* subselection mechanism, analogous to that presented in Roth et al. (2022); Sener & Savarese (2018); Sinha et al. (2020). The process is applied separately to the class-specific memory banks to obtain subsampled versions $\hat{\mathcal{M}}_{y=1}^{|\mathcal{Y}^s|}$, which are then aggregated in a unified bank $\hat{\mathcal{M}} = \bigcup_{y=1}^{|\mathcal{Y}^s|} \hat{\mathcal{M}}_y$

## 3.3 SCORING FUNCTION

For each test sample $x^t$ we start by extracting the set of patch embeddings $\phi_l(\boldsymbol{x}^t) = \{\boldsymbol{v}_k\}_{k=1,\dots,P_l}$. For each patch we compute the following two properties by simple nearest neighbor matching with the memory bank:

$$\delta(\boldsymbol{v}_k) = \min_{\hat{\boldsymbol{v}} \in \hat{\mathcal{M}}} d(\boldsymbol{v}_k, \hat{\boldsymbol{v}}) \tag{1}$$

$$\lambda(\boldsymbol{v}_k) = y_{v^*} \mid v^* = \arg\min_{\hat{\boldsymbol{v}} \in \hat{\mathcal{M}}} d(\boldsymbol{v}_k, \hat{\boldsymbol{v}}) . \tag{2}$$

They respectively represent the distance from the memory nearest patch and the corresponding class assignment. For a test sample the class assignments probability is $P_{x^t}(y) = (1/P_l) \sum_{k=1}^{P_l} \mathbb{1}_{\lambda(v_k)=y}$, and its normality score can be computed as the inverse entropy of the class probabilities:

$$H = \mathbb{E}_{x_t}[\log(P_{x_t}(\lambda(v)))] . \tag{3}$$

This will yield low normality scores when the assignments are disordered and belong to many different classes. The intuition is correct but the formulation ignores the embedding distances which could convey useful information about the novelty of the sample. Thus, we enhance it by weighting the entropy with the patch distances:

$$H_w = \mathbb{E}_{x_t}[\delta(v) \log(P_{x_t}(\lambda(v)))] . \tag{4}$$

In this way we solve the ambiguity arising when OOD samples are composed of patches that have a high embedding distance from the memory bank patches but consistently match with a limited set of patches belonging to the same class, resulting in low entropy.

## 4 EXPERIMENTS

Simple methods that build on large pre-trained models are particularly appealing for complex application scenarios. Besides their accuracy, the value of these methods should be assessed considering aspects such as the robustness to the specific pre-training in terms of backbone and learning objective, as well as their sample efficiency. For OpenPatch we ran an extensive evaluation on these aspects. We focus on the challenging Synthetic to Real benchmark from 3DOS Alliegro et al. (2022). Further evaluations on other tracks and different pre-trainings can be found in the supplementary material.

### 4.1 EXPERIMENTAL SETUP

**Pre-training.** We consider two different large-scale 3D models as starting point for OpenPatch: a single-modal one trained on Objaverse Deitke et al. (2023), and a multi-modal one named Open-Shape Liu et al. (2023) trained on the combination of Objaverse and other three 3D datasets and related text-descriptions.

Objaverse is a dataset of annotated 3D shapes collected from free internet resources. We use the subset proposed by the authors called *Objaverse-LVIS* Deitke et al. (2023) that contains point clouds divided into 1156 semantic classes and 47K samples. The classes reflect the ones proposed in the LVIS dataset Gupta et al. (2019) and their sample assignments were obtained using CLIP and the objects' metadata. We trained two different backbones with object classification objective. *PointNet++* Qi et al. (2017) is a widely recognized and commonly used architecture in the realm of 3D learning. *EPN* Chen et al. (2021) is well-suited for real-world settings, where the pose of the test object is not known beforehand. In the training process we augment training data with jittering, SO(3) rotation, random rescaling, random translation, and random crop of a small neighborhood of points. For OpenPatch we need to focus on a specific network layer to extract the patch embeddings. We utilize the 4th convolutional block of the EPN backbone obtaining $\boldsymbol{v}_k \in \mathbb{R}^{64}$, and the 2nd convolutional block of the PointNet++ backbone obtaining $\boldsymbol{v}_k \in \mathbb{R}^{128}$.

For OpenShape we adopt the models provided by the authors, trained via multi-modal contrastive learning for representation alignment. Specifically, we consider two different backbone architectures: PointBert Yu et al. (2022) and SPConv Choy et al. (2019). The former is a transformer-based backbone inspired by BERT Devlin et al. (2018), and the latter is a CNN using sparse convolutions. The patch embedding for OpenPatch are extracted respectively from the last transformer block and the last sparse convolutional layer.

**3DOS.** 3DOS leverages the ShapeNetCore Chang et al. (2015), ModelNet40 Wu et al. (2015), and ScanObjectNN Uy et al. (2019) datasets to provide a vast array of benchmark tracks, we focus on the Synth to Real track as it is the most realistic and difficult setting. This track involves a support set of synthetic point clouds from ModelNet40 Wu et al. (2015) and a test set of real-world point clouds from ScanObjectNN Uy et al. (2019). Three category sets, SR1, SR2, and SR3, are defined. SR1 and SR2 consist of matching classes from ModelNet40 and ScanObjectNN, while SR3 includes ScanObjectNN classes without a one-to-one mapping with ModelNet40. Two scenarios of increasing difficulty are considered, where either SR1 or SR2 is used as the known class set, and the other two sets are considered unknown.

**Reference Methods.** Most of the efforts dedicated to the design and training of large scale models aim at learning a rich and reliable representation, reusable for several downstream tasks. The basic assumption is that the internal network features capture relevant semantic knowledge and that simple metric relations among points in the learned high-dimensional embedding provide discriminative information. Thus, the simplest way to probe a representation space is by using nearest neighbors. For OOD detection, this means defining the normality score for a test sample as the inverse of the Euclidean distance in the feature space between the embedded test sample and its nearest neighbor within the support set. We indicate this as *1NN*. Besides this non-parametric solution, alternative techniques estimate per-class feature distributions: *EVM* assumes the embeddings to conform to a Weibull distribution while *Mahalanobis* assumes a multivariate Gaussian distribution. At test time the first computes the likelihood of the test sample for each class-distribution and uses the maximum likelihood as the normality score, while the second uses the Mahalanobis distance from the closest class distribution as the normality score. As it is clear, all these approaches do not need any learning stage on the ID support set for OOD detection, thus they are fair competitors of OpenPatch.

To position OpenPatch in the standard OOD detection literature we also consider in our analysis three post-hoc methods based on a classifier trained on the support set. *MSP* Hendrycks & Gimpel (2017) exploits the maximum softmax probability as a normality score, assuming that unknown samples will be classified with lower confidence. *MLS* Vaze et al. (2022) proposes to discard the normalization step provided by the softmax application, and uses the maximum logit value directly. *ReAct* Sun et al. (2021) improves the known-unknown separation by applying a rectification on the network activations.

**Performance metrics.** As the semantic novelty detection problem is inherently a binary task we employ *AUROC* and *FPR95* Hendrycks & Gimpel (2017) as evaluation metrics. We use the term *positive* for nominal samples and the term *negative* for the unknown ones. The AUROC (higher is better) is the Area Under the Receiver Operating Characteristics curve, which plots the True Positive Rate against the False Positive Rate when varying a threshold applied to the predicted positive scores. As a result, this metric is threshold-independent and can be interpreted as the probability for a nominal sample to have a greater score than an unknown one. The FPR95 (lower is better) is the False Positive Rate computed when the threshold is set at the value that corresponds to a True Positive Rate of 95%. Although AUROC is the metric that better reflects the potential ability of a method to correctly detect novelty, FPR95 provides a more concrete idea of the operational performance of an OOD method as it provides a guarantee about the safe recognition of known data and gauges the risk of mistakenly accept as known a sample of an unseen object class.

## 4.2 RESULTS

**Pre-training and fine-tuning free OOD detection benchmark.** Our first set of experiments is dedicated to a comparison of the OOD detection performance of OpenPatch with that of the other fine-tuning free methods 1NN, EVM and Mahalanobis when starting from the internal representation of several pre-trained networks that differ in terms of used data and learning objective.

| Pre-train Objaverse-LVIS | EPN Chen et al. (2021) | | | | | | PointNet++ Qi et al. (2017) | | | | | |
|---|---|---|---|---|---|---|---|---|---|---|---|---|
| | SR1 (easy) | | SR2 (hard) | | Avg | | SR 1 (easy) | | SR 2 (hard) | | Avg | |
| *Method* | AUROC↑ | FPR95↓ | AUROC↑ | FPR95↓ | AUROC↑ | FPR95↓ | AUROC↑ | FPR95↓ | AUROC↑ | FPR95↓ | AUROC↑ | FPR95↓ |
| 1NN | 57.5 | 95.1 | 56.8 | 90.2 | 57.3 | 92.6 | 60.1 | 90.7 | 57.5 | 87.8 | 58.8 | 89.3 |
| EVM Rudd et al. (2015) | 71.6 | 80.0 | 60.6 | 91.5 | 66.1 | 85.7 | 60.7 | 92.2 | 61.3 | 87.7 | 61.0 | 90.0 |
| Mahalanobis Lee et al. (2018) | 61.1 | 92.0 | 60.2 | 91.2 | 60.6 | 91.6 | 61.2 | 92.5 | 58.2 | 85.7 | 59.7 | 89.1 |
| OpenPatch | 71.9 | 83.6 | 72.1 | 79.2 | **72.0** | **81.4** | 69.9 | 86.7 | 63.3 | 92.2 | **66.6** | **89.5** |

Table 1: Results on the Synth to Real Experiments for the EPN and Pointnet++ backbones trained on Objaverse-LVIS dataset.

| Pre-train OpenShape | SPCONV Choy et al. (2019) | | | | | | PointBert Yu et al. (2022) | | | | | |
|---|---|---|---|---|---|---|---|---|---|---|---|---|
| | SR1 (easy) | | SR2 (hard) | | Avg | | SR 1 (easy) | | SR 2 (hard) | | Avg | |
| *Method* | AUROC↑ | FPR95↓ | AUROC↑ | FPR95↓ | AUROC↑ | FPR95↓ | AUROC↑ | FPR95↓ | AUROC↑ | FPR95↓ | AUROC↑ | FPR95↓ |
| 1NN | 82.4 | 71.9 | 61.6 | 94.5 | 72.0 | 83.3 | 73.4 | 85.8 | 58.9 | 87.7 | 66.2 | 86.7 |
| EVM Rudd et al. (2015) | 76.0 | 85.1 | 59.6 | 95.5 | 67.8 | 90.3 | 76.0 | 88.6 | 58.9 | 93.5 | 67.5 | 91.1 |
| Mahalanobis Lee et al. (2018) | 62.6 | 93.4 | 54.2 | 92.8 | 58.4 | 93.13 | 73.8 | 89.7 | 65.3 | 83.3 | 69.5 | 86.5 |
| OpenPatch | 85.7 | 68.7 | 73.2 | 80.1 | **79.4** | **74.4** | 85.8 | 54.4 | 71.6 | 74.1 | **78.7** | **64.3** |

Table 2: Results on the Synth to Real Experiments for the SPCONV and PointBert backbones trained with OpenShape's method.

Results in Tab. 1 reveal that OpenPatch surpasses all competitors in both SR1 and SR2 tracks by a large margin. This observation underscores the inadequacy of relying solely on distance-based techniques: a more structured solution, such as that provided by OpenPatch, which is able to capture local and global information about the 3D objects at hand, is essential for semantic novelty detection. Furthermore, through the comparison of EPN and PointNet++ backbones we demonstrate that the representation obtained by adopting a rotation-invariant backbone is particularly suited for real-world semantic novelty detection. As in our experiments the support set is composed of synthetic data and the test set contains real point clouds there is no control on the sample orientation and EPN leads to a substantial performance improvement, of more than 5 AUROC points, over PointNet++. Minor improvements are also observed for competitors.

Tab. 2 shows the results obtained when starting from the OpenShape Liu et al. (2023) pre-trained multimodal embedding. The increased volume and multi-modality of the data at the basis of the OpenShape learned representation clearly yield remarkably strong performance. Still, even in this case OpenPatch wisely exploits this strong feature encoding and overcomes its competitors. In comparison to the 1NN baseline, OpenPatch showcases a large enhancement of 7.4 points in AUROC when employing the SPCONV backbone, and an even more remarkable improvement of 12.5 AUROC points when utilizing the PointBert backbone. These findings underscore the versatility and adaptability of OpenPatch, showcasing its ability to leverage feature encoding quality enhancements for improved results.

**Sample Efficiency.** As previously stated, methods that avoid training on the ID support set represent a preferable choice for many applications in which a learning phase may be unfeasible for various reasons as limited computational capacity and scarce data availability. To assess OpenPatch for sample efficiency we introduce a few-shot experimental setting that reflects the realism of industrial scenarios, where obtaining large-scale collections of accurate CAD files for known objects is often impractical. For each benchmark track, we create splits with K randomly selected samples from each class in the support set with $K \in \{5, 10, 20, 50\}$. We repeat the sampling process 10 times for each $K$, and report in Figure 2 the average results.

In particular, we evaluate the few-shot performance of the leading backbones, which are EPN from Table 1 and PointBert from Table 2, across different methods. For both backbones, OpenPatch outperforms all other competitors and retains stable results even at low K values.

**Training on the Support set.** We present a comparative analysis to confidence-based OOD detection methods to better position OpenPatch in the OOD detection literature where standard models are directly trained or fine-tuned for classification on the ID samples of the support set. We adopt the EPN backbone and for the finetuning-variant we start from the Objaverse-LVIS pre-training. The results in Fig. 2 (right) indicate that when learning on the support set is possible it is clearly advantageous in terms of AUROC, with ReAct showing top performance. Still the AUROC of OpenPatch (72.0) is comparable to that of MSP (71.6) and MLS (72.3) without pre-training, as well as MSP (72.8) with pre-training. Remarkably, OpenPatch shows the overall top performance in terms of FPR95: this is

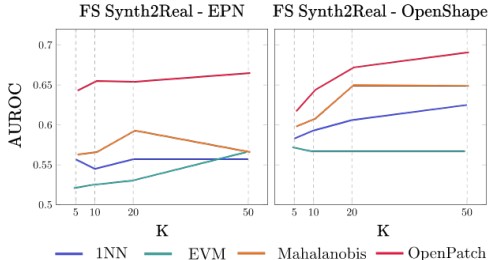

| Train on the Support Set | EPNChen et al. (2021) | | | | | |
|---|---|---|---|---|---|---|
| | SR1 | | SR2 | | Avg | |
| *Method* | AUROC↑ | FPR95↓ | AUROC↑ | FPR95↓ | AUROC↑ | FPR95↓ |
| Methods trained from scratch on the support set (ID) | | | | | | |
| MSP Hendrycks & Gimpel (2017) | 74.0 | 89.1 | 69.1 | 89.8 | 71.6 | 89.5 |
| MLS Vaze et al. (2022) | 72.8 | 92.8 | 71.7 | 79.0 | 72.3 | 85.9 |
| ReAct Sun et al. (2021) | 76.6 | 92.5 | 72.2 | 76.7 | **74.4** | 84.6 |
| Methods fine-tuned on the support-set (ID) | | | | | | |
| MSP Hendrycks & Gimpel (2017) | 74.3 | 88.5 | 71.3 | 83.4 | 72.8 | 85.9 |
| MLS Vaze et al. (2022) | 72.8 | 87.7 | 73.4 | 79.9 | 73.1 | 83.8 |
| ReAct Sun et al. (2021) | 73.6 | 90.4 | 73.9 | 76.1 | **73.7** | 83.2 |
| OpenPatch | 71.9 | 83.6 | 72.1 | 79.2 | 72.0 | **81.4** |

Figure 2: Left: Few-Shot experiments on the Synth to Real track, K represents the number of samples in the support set for each class. Right: Results on the Synth to Real benchmark when training from scratch or fine-tuning on the support set (ID) starting from the Objaverse-LVIS pre-training.

particularly interesting as this metric indicates the operational effectiveness of OpenPatch when the accuracy on the recognition of the ID data is guaranteed to be over 95%. As a side note, we highlight how fine-tuning may degrade the performance: in particular ReAct suffers from a minimal AUROC loss. finally, it is worth emphasizing that having access to a large number of ID samples, which is essential for network optimization, is often unfeasible in industrial or real-world scenarios - the very contexts where OpenPatch demonstrates its effectiveness.

### 4.3 COMPONENT ANALYSIS

As described in Sec 3, OpenPatch presents three main components: the patch feature extraction, the memory bank with the related coreset subsampling, and the scoring function. For each of them it is possible to operate different choices. We provide here a comprehensive analysis of the behavior of OpenPatch with the EPN backbone when modifying each of the components.

**Extraction Layer.** The first plot in the left part of Figure 3 shows that the performance of OpenPatch grows when moving from a shallow layer (close to the network input) towards a deeper one. Extracting patch embeddings at deeper layers provides the added benefit of generating fewer feature vectors per sample. Consequently, we can apply a less aggressive coreset reduction and maintain a short evaluation time.

**Coreset Reduction.** The second plot in the left part of Figure 3 shows how much the performance of OpenPatch is sensitive to the chosen coreset dimension. We can see that even when reducing it by 1/5 the performances remain stable, after that the performances quickly deteriorate.

Both the extraction layer choice and the coreset reduction are key for the deployment in memory-constrained environments. Using deeper layers both improves performance and reduces the cardinality of the memory bank. The coreset sampling technique offers us an advantageous trade-off between memory consumption and performance.

**Scoring functions.** The table in the right part of Figure 3 presents the results obtained by changing the scoring function used to evaluate whether a test sample belongs to the known classes. We compare also with the simple baselines of max and mean functions expressed in the following way: $Max = max_{k=1,...,P_l}(\delta(\boldsymbol{v}_k))$ and $Mean = \sum_k^{P_l}(\delta(\boldsymbol{v}_k))/P_l$. These experiments highlight that our tailored scoring function together with the choice of the extraction layer define the major strength of our method. together with the choice of the extraction layer. The results confirm our choice intuition regarding entropy-based scorings: overall both Entropy ($H$) and Weighted entropy ($H\_w$) perform significantly better than simple distance-based scorings.

### 4.4 QUALITATIVE ANALYSIS

A significant challenge in OOD detection involves unraveling the rationale behind classifying a particular sample as either known or unknown, as well as identifying the part of the object that contributed to that decision. Diverging from the approach of the fine-tuning-free competitors, which generate a global prediction for each object, OpenPatch stands out by operating at a finer level of granularity, specifically at the level of object patches. This design choice enhances the capacity of OpenPatch to provide a more nuanced interpretation of the final OOD detection outcome. In Fig. 4

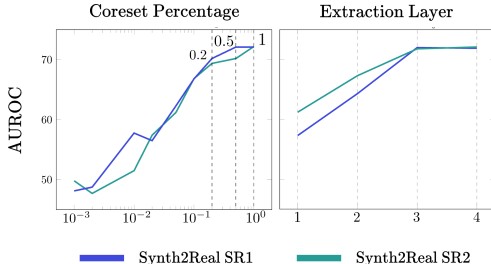

| Scoring | Synth to Real SR1 | | Synth to Real SR2 | | Avg | |
|---|---|---|---|---|---|---|
| Functions | AUROC↑ | FPR95↓ | AUROC↑ | FPR95↓ | AUROC↑ | FPR95↓ |
| Max | 55.2 | 93.6 | 60.9 | 92.7 | 58.1 | 93.2 |
| Mean | 63.5 | 82.3 | 68.3 | 84.9 | 66.6 | 83.6 |
| $H$ | 72.0 | 85.1 | 70.8 | 86.8 | 71.4 | 85.9 |
| $H_w$ | 71.9 | 83.6 | 72.1 | 79.2 | **72.0** | **81.4** |

Figure 3: Left: AUROC trends of OpenPatch when varying the *Coreset Percentage* and the *Extraction Layer*. Right: OpenPatch with different *Scoring Functions* on the 3DOS Synth to Real tracks Experiments are performed with EPN Chen et al. (2021) backbone.

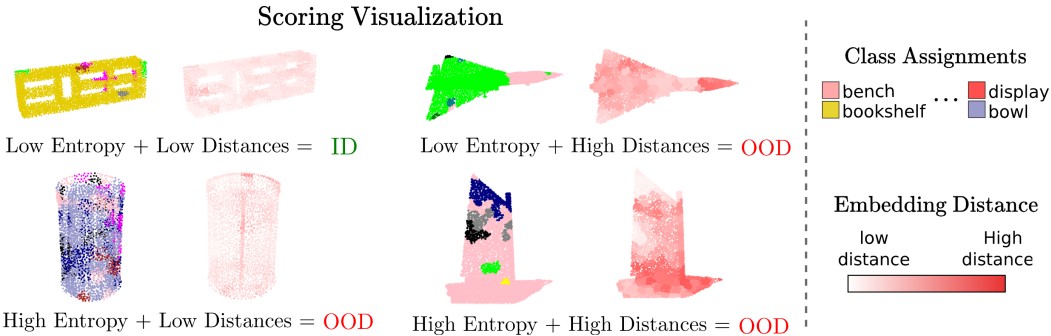

Figure 4: Qualitative evaluation of OpenPatch with PointNet++ backbone and interpretation of the scoring function, for each sample the left visualization represents the class assignments, while the right visualization represents the embedding distance

we report visualizations of patch extraction and interpretations of our scoring function. Both class assignments (left) ad embedding distance (right) are color-coded, each point inherits the color from the closest patch. From the visualizations we can clearly distinguish when a sample is considered as belonging to a novel category and why, if the network is mixing object classes or doesn't recognize parts of the original object it will probably be classified as unknown.

## 5 CONCLUSION

We introduced OpenPatch, a model that effectively detects semantic novelty in 3D data without requiring to be trained on support set ID data. We proposed a strategy to extract generalizable patch features from a pre-trained 3D deep learning architecture and we designed an innovative approach that combines semantic and relative distance information to accurately identify samples belonging to novel classes during testing.

What sets OpenPatch apart is its remarkable ability to operate in limited data scenarios and its resistance to domain shifts regarding the Synth to Real scenario. OpenPatch proves to be a flexible solution for real-world applications and provides clear and intuitive visualization to understand its inner functioning. It can be effectively deployed in data-constrained environments, eliminating the need for collecting custom data collections and training task-specific models.

## 6 REPRODUCIBILITY

We provide in section 4.1 the most relevant information on the dataset and implementations of our method. In the supplementary material we describe all the training details and the hardware setup adopted for our whole experimental analysis. For OpenShape experiments we use the checkpoints publicly available at their repository. Our code implementation will be publicly released upon acceptance.

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
