# OpenReview forum: "OpenPatch: a 3D patchwork for Out-Of-Distribution detection"
_ICLR.cc/2024/Conference — Submitted to ICLR 2024_

### Official Review · Reviewer_3HQY · 2023-10-31

**Soundness:** 3 good
**Presentation:** 3 good
**Contribution:** 2 fair
**Rating:** 6
**Confidence:** 2

**Summary:**

In this paper, the authors propose a 3D patchwork for out-of-distribution detection. They propose to use the intermediate patch features from a pre-trained 3D convolutional network to create a memory bank for point cloud patches. The authors conduct experiments on various point cloud backbones to demonstrate the effectiveness of the proposed OpenPatch. Ablation studies are also provided for component analysis.

**Strengths:**

1. The idea is novel. The intermediate features from the pretrained point cloud model is adequately used to solve the OOD problem.
2. The visualization is clear. The quatitative evaluation of OpenPatch in Figure 4 is clear to demonstrate the concept and the effectiveness of the proposed method.

**Weaknesses:**

1. In the experiment part, the authors only compare OpenPatch with a few related literatures. However, from section 2, there are many related work that are not included in the experiment part. The authors are encouraged to provide more robust experiment comparison to convince the effectiveness of the proposed method.
2. There are more advanced backbones in point cloud representation learning field. The authors are encouranged to show that OpenPatch also works well on more advanced backbones.
3. In figure 4, the concept of "Patch" is not clearly demonstrated. It would be nice the the authors can show the "Patch"-level visualizations.

**Questions:**

See weakness above.

---

> ### Author Response · Authors · 2023-11-23
>
> We thank the reviewer for the positive feedback.
>
> **W1**: Most of the OOD detection literature covered by our sec 2 is tailored to 2D data and considers methods that require a training or at least a finetuning phase on the support set that contains samples of known classes. Our approach is instead learning-free. OpenPatch exploits a pre-trained architecture to extract data representations and perform sample-to-class comparisons to detect novelty in 3D data. Despite the different scenarios, the right part of Fig. 2 shows that OpenPatch is competitive with the state-of-the-art in 3D OOD detection methods (according to [Alliegro et al. (2022)]) that train or finetune on the support set.
>
> **W2**: Unfortunately the reviewer did not provide pointers to specific advanced backbones. We used both well-known architectures on Objaverse and cutting-edge transformer-based ones on OpenShape (models released with the publication [Liu et al. (2023)]).
>
> **W3**: In Figure 4, we present a depiction of patch assignments using a backbone based on Farthest Point Sampling (FPS). For this instance, we extracted the coordinates of the patch features and subsequently assigned each point in the original point cloud to its nearest patch feature. In the initial column, we color-coded based on the class assignment of the underlying patch embedding, while in the second column, each point was colored using heatmap colors corresponding to the relative embedding distance from the memory bank of the underlying patch embedding.
> We recognize that discerning the different patches could be difficult due to the color similarity, to counteract this, we added an additional figure in the supplementary material where we color each patch a different color and compare the patch dimension when changing the layer of extraction.

---

### Official Review · Reviewer_shas · 2023-11-01

**Soundness:** 3 good
**Presentation:** 3 good
**Contribution:** 2 fair
**Rating:** 3
**Confidence:** 3

**Summary:**

This paper aims to identify novel classes in 3D point clouds. It uses intermediate features to distinguish known semantic classes from novel classes that are not seen during training. It stores patch embeddings in a memory bank, and proposes a score function to find novel classes. Experiments are performed with full and few-shot scenarios.

**Strengths:**

- Detection of novel classes is indeed an important task. While many methods have been proposed in 2D, it has been less explored in 3D applications.
- The paper is generally clear and easy to understand

**Weaknesses:**

- The task of out-of-distribution detection is closely related to the open-set and open-world task. Nonetheless, the paper does not compare to methods that perform open-world tasks.
"Walter J Scheirer, Anderson de Rezende Rocha, Archana Sapkota, and Terrance E Boult. Toward open set recognition. IEEE transactions on pattern analysis and machine intelligence (2012)"
"Abhijit Bendale and Terrance Boult. Towards open world recognition. In Proceedings of the IEEE conference on computer vision and pattern recognition (2015)"

- The paper is mainly a simple adaptation of what has been tried in 2D (memory bank and scoring) and falls short in terms of originality. Moreover, it does not compare to other out-of-distribution methods from 2D.
- The motivation for a 3D framework is somehow weak since the paper does not explain the challenges in the 3D domain compared to the direct adaptation from 2D.
- There no proper ablation studies.
- Using "detection" in the title is confusing since the paper is only doing classification and semantic segmentation.

**Questions:**

- How is out-of-distribution detection different from open-set and open-world?
- What are the challenges/differences in the 3D domain compared 2D?

---

> ### Author Response · Authors · 2023-11-23
>
> We thank the reviewer for the feedback. The questions highlight a potential gap in familiarity with the existing literature, which we are happy to address through our responses.
>
> **W1 + Q1**: Out of Distribution Detection aims at labeling a new sample as belonging or not to the training distribution. This is a binary problem. Open Set Recognition refers instead to a class set, so its goal is identifying if a new sample belongs to one of the known classes or to an unknown one. This is a multiclass problem and more precisely, it combines the multiclass recognition of the known classes with the binary task of separating known from unknown data. Open World builds over Open Set, so that once a new class is identified it is incrementally added to the original model.
> Our focus is on Out of Distribution Detection since we are only interested in the binary task of separating known from unknown data. As in our specific case the unknown data is novel because it belongs to a new semantic class, we also used the expression ‘semantic novelty detection’.
> We are aware that the names might be confusing at first sight, but they have been explained in the first paragraph of our sec. 2.
> Having clarified the frameworks it also becomes clear that the papers cited by the reviewers do not match our setting.
>
> **W2**: We use memory banks but we did not claim novelty on it. Our technical contribution is in the 3D-tailored patch extraction and OOD scoring function. We are the first to propose a metric for 3D OOD detection that combines semantic consistency and patch anomalousness.
> Moreover, our approach does not need training on the support dataset of known classes, while most of the OOD detection literature relies on this stage. This makes a direct comparison unfair. The approaches that we considered as baseline are the only methods that afford OOD detection without learning/finetuning on the support set. See also the answer to W1 from the reviewer 3nE.
>
> **W3 + Q2**: The work [Alliegro et al. (2022)] extensively discussed the different challenges of 3D OOD detection and Open Set Learning with respect to those of the corresponding 2D domain.
>
> **W4**: The component analysis presented in section 4.3 explains in detail the role of each component of our approach. We would be happy to provide further details if the reviewer could specify the meaning of ‘proper ablation’ for our model.
>
> **W5**: Out of Distribution Detection is a standard name to indicate the task of predicting whether a test sample belongs or not to the nominal data distribution. Unfortunately there is a clear misunderstanding, we are not performing neither object detection nor semantic segmentation.

---

### Official Review · Reviewer_RMdW · 2023-11-03

**Soundness:** 1 poor
**Presentation:** 1 poor
**Contribution:** 1 poor
**Rating:** 3
**Confidence:** 4

**Summary:**

This paper proposes a new method called OpenPatch for out-of-distribution (OOD) detection on 3D point clouds. The authors use a pre-trained 3D convolutional neural network to extract semantically and geometrically meaningful patch embeddings from intermediate layers. These patches represent local parts of an object. They then build class-specific memory banks containing patch embeddings extracted from known in-distribution samples. Apply coreset subsampling to reduce redundancy.  For testing, the authors propose a scoring function that extracts patch embeddings and find nearest neighbors in the memory banks. Evaluate novelty based on 1) distance to nearest patches and 2) entropy of class assignments. High entropy indicates the sample matches patches from many classes, suggesting it is OOD. Experiments show OpenPatch outperforms distance-based methods like nearest neighbors, as well as post-hoc methods that train classifiers on in-distribution data. It is also sample efficient, performing well even with few support examples per class.

**Strengths:**

Not much could be said here.

**Weaknesses:**

1. The importance of determining if the point cloud of an object is out-of-domain for a 3d classifier is not well-presented. No prior works nor possible applications are shown for this work.
2. The authors proposed a patch-based method for determining if a point cloud is OOD but failed to mention how are those patches obtained. I guess how to break a whole point cloud into patches has a significant impact on the OOD recognition accuracy. In this sense, the proposed method is not even complete.
3. Extracting patch-wise features with deep networks is nothing new and the authors did not work hard on literature reviews. I am supervised the authors failed to discuss the relation between this work and deep VLAD approaches like NetVLAD and PointNetVLAD. Also patch-wise methods like BoW with deep CNNs back in 2015 are highly related to this work.

**Questions:**

The authors should spend more time studying related works BoW and VLAD with deep features back in the earliest days of deep learning.

---

> ### Author Response · Authors · 2023-11-23
>
> We thank the reviewer for the comments. Unfortunately, a few observations might be based on misunderstandings. We offer some clarification in the following.
>
> **W1**: Regarding the “no prior work” claim, we kindly refer the reviewer to [Alliegro et al (2022)] in our references. This work introduced the task of 3D OOD detection and ran an extensive benchmark of methods originally designed for 2D data, showing that their performance does not necessarily transfer to 3D.
> Regarding the “no possible application” claim, we highlight the essential need for methods able to manage novelty in real-world applications involving 3D information. We refer the reviewer to the following works that appear among our references [Masuda et al. (2021); Bhardwaj et al. (2021); Cen et al. (2021); Wong et al. (2019); Riz et al. (2023)]. Besides autonomous driving where unexpected objects might create critical safety issues, other relevant scenarios are related to industrial robotics applications where agents are supplied with only a few clean 3D object templates to use as reference and they should be able to mitigate potential operational hazards due to novel objects.
> Both prior works and applications are mentioned in the paper's introduction and related work section.
>
> **W2**: Regarding “failed to mention how are those patches obtained” we point the reviewer to sec 3 (in particular 3.1) of our submission. There, we explained that representations extracted from internal layers of networks that elaborate on point clouds describe different sized 3D shape portions that we indicate as patches. There is no manual segmentation involved here, the patches are a byproduct of the chosen architecture and the only choice to operate is deciding from which layer to extract them: deeper layers provide larger patches.
>
> **W3 +  Q**: Both BOW and VLAD define global representations from local descriptors to compare samples among each other. Specifically, NetVLAD is an architecture designed and trained (with specific loss) to learn such a representation. The goal is then to use it to search for the most similar samples of a certain query instance in a retrieval scenario.
> Differently, our work proposes a strategy to extract from large-scale pre-trained models (regardless of the specific backbone and loss) a representation that can be efficiently used to evaluate whether a sample belongs or not to a reference class via a newly introduced scoring function and no further training. Incidentally, collecting features from different layers of a point-cloud network means collecting local representations, thus we designed the metric to evaluate the assignment to known or unknown class to exploit this logic.
> We remark that we care about sample-to-class distance rather than sample-to-sample distance. It is well known that BOW-related strategies are suited for the latter but not for the former scenario [*]
>
> [*] “In Defense of Nearest-Neighbor Based Image Classification”, CVPR 2008

---

### Official Review · Reviewer_3nEL · 2023-11-04

**Soundness:** 1 poor
**Presentation:** 1 poor
**Contribution:** 2 fair
**Rating:** 3
**Confidence:** 4

**Summary:**

This paper presents a novel approach to Out-Of-Distribution (OOD) detection in the realm of 3D data, termed OpenPatch. It addresses the challenge of detecting novel objects (OOD data) that a deep learning model has not encountered during its training phase. The core contributions of the paper can be summarized as follows:

(1) Development of OpenPatch: OpenPatch, a novel OOD detection method, leverages patch representations from intermediate layers of pre-trained 3D models. This approach enables the detection of novel 3D objects by differentiating between known and unknown object categories without the need for any additional training, addressing the limitations of previous methods that rely heavily on fine-tuning with extensive support data sets.

(2)  Plug-and-Play Capability with Semantic Novelty Detection: OpenPatch operates as a plug-and-play solution for real-world applications, notably in industrial robotics, where computational resources and available data are limited. It is designed to detect semantic novelty effectively by comparing the test sample’s patch features against a database of known classes, using distance metrics and entropy-based class assignment diversity to identify OOD instances, thereby mitigating potential operational hazards.

(3) Demonstrated Superiority and Efficiency: The approach not only surpasses existing OOD detection methods in performance but also showcases high sample efficiency and resilience to domain bias. This makes OpenPatch highly suitable for practical applications, as it can be readily deployed without retraining for different tasks or updating the nominal support set, which is a significant advancement over existing 2D and 3D OOD detection techniques.

In summary, the paper introduces a robust and efficient method for detecting new objects in 3D data, which is of particular relevance for industrial applications and other real-world scenarios where data and computational resources are constrained.

**Strengths:**

OpenPatch's strengths are concentrated in the following areas:

(1) Novelty in distinguishing known and unknown classes: OpenPatch introduces a strategy for extracting generalizable patch features from pre-trained 3D deep learning architectures. It devises an innovative approach that integrates semantic and relative distance information to accurately identify new categories during the testing phase.

(2) Streamlined deployment: OpenPatch can be efficiently deployed in resource-constrained environments, obviating the need for collecting custom datasets and training task-specific models.

**Weaknesses:**

(1) About innovation.
The approach taken in this paper is too similar to PatchCore. For example, the "Patch Feature Extractor" in OpenPatch mirrors the "Local-Aware Patch Feature" in PatchCore, while the "Memory Bank and Subsampling" in OpenPatch is very similar to the "Coreset Reduced Patch Feature Memory Bank" in PatchCore. And the key strategies selected by the two papers are similar, such as Greedy Coreset and KNN. Overall, I think this paper lacks innovation.

(2) About experiments.
A. The selected comparison methods are out of date. For example, in Tables 1 and 2 the chosen comparison methods are EVM Rudd et al. (2015) and Mahalanobis Lee et al. (2018), have become obsolete.
B. The experimental advantages are not obvious. For example, in the "Training on Support Set" table, only 3 out of 6 main metrics outperform previous methods.

(3) About writing.
The paper requires improvement in both writing and logic, particularly in the abstract and introduction sections.

**Questions:**

(1) About experiments.

Table 1 and Table 2 demonstrate the performance advantages of OpenPatch on OOD compared with other methods. Moreover, the experiments in Table1 are pre-trained on Objaverse-LVIS dataset, while Table 2 shows the results obtained when starting from the OpenShape pre-trained multimodal embedding. But I have some questions about the choices of comparison methods:

A.In Table 1 and Table 2, compared with methods such as EVM and Mahalanobis that have been proposed for many years, if the latest research such as [1][2][3] are added to the comparison, can OpenPatch maintain its advantage?

[1]Semantic Novelty Detection via Relational Reasoning

[2]Detecting out-of-distribution examples with Gram matrices

[3]Delving into Out-of-Distribution Detection with Vision-Language Representations

B.If the latest research [4] and [5] are introduced into the backbone selection of Table 2, what changes will occur in the experimental results?

[4]Uni3D: Exploring Unified 3D Representation at Scale

[5]ViT-Lens: Towards Omni-modal Representations


(2) As mentioned in 3.2, "The banks" cardinality may quickly increase, significantly impacting the computational cost of the method. To mitigate this effect and address redundancy we adopt a greedy coreset subselection mechanism.” Can the impact of greedy coreset subselection mechanism on computational cost reduction be quantitatively reflected in the experiment?

---

> ### Author Response · Authors · 2023-11-23
>
> We thank the reviewer for the detailed feedback.
>
> **W1**: Regarding the similarities with Patchcore, we inherit its memory bank creation and downsampling mechanism. We did not claim novelty on either of them. Our technical contribution is in the 3D-tailored patch extraction and OOD scoring function. We are the first to propose a metric for 3D OOD detection that combines semantic consistency and patch anomalousness.
>
> **Q1 A**: Concerning the mentioned references we can state that:
> [1] leverages ad hoc pre-training and a very specific architecture designed to be used for semantic novelty detection on 2D data. Instead, our work targets efficiency in design and learning effort: OpenPatch builds on large-scale 3D models trained with standard classification or contrastive objectives and well-known backbones.
> We remark that, despite 3D transformer models exist, adapting the relational module and the learning pipeline of [1] to the 3D setting is not straightforward.
> The method proposed in [2] was shown in [1] to be worse than the Mahalanobis baseline that we adopted. Therefore, although adding it to our analysis might improve completeness, our conclusions would remain unchanged.
> The approach in [3] requires the availability of a textual description for the known data object category. This cannot be given for granted in industrial settings where in-distribution objects are custom CAD models created by users and are possibly identified by numerical codes without any semantic meaning.
>
> **Q1 B**: [4] and [5] propose scaled-up architectures and improved multi-modality training over OpenShape but they maintain the contrastive loss paradigm. In other words, they learn better 3D representation from which our OpenPatch would benefit: we expect linear improvements over the reported results without any significant change in the general trend.
> Additionally, [4] is a concurrent ICLR submission we were not aware of while working on OpenPatch, while [5] is an arXiv paper.
>
> **Q2**: The main bottleneck of OpenPath lies in the nearest neighbor search which depends on the number of patches stored in the memory banks. This total count of patches is determined by the number of samples in the support set and the layer used for feature extraction in the pretrained model encoder. In our specific experiments, we observed negligible performance gains from implementing a more aggressive coreset reduction. We attribute this to the low number of samples in the support set, as defined by the 3DOS benchmark, and our decision to use larger patches rather than numerous smaller local patches. Our main concern for implementing the coreset selection is mainly the limited memory of devices in industrial scenarios.

---

### Meta-Review · Area_Chair_2Dmx · 2023-12-04

**Metareview:**

Dear authors,

Thank you for submitting the draft, on an interesting problem. Unfortunately, majority of the authors have indicated that this draft is not ready for the publication. We encourage authors to take inconsideration reviewers' comments when updating the draft, especially clearly presenting their contribution and differentiating those from previous works.

regards
Meta-reviewer

**Justification For Why Not Higher Score:**

Majority of reviews are not favorable.

**Justification For Why Not Lower Score:**

N/A

---

### Decision · Program_Chairs · 2024-01-16

Reject